# Association between Oncostatin M Expression and Inflammatory Phenotype in Experimental Arthritis Models and Osteoarthritis Patients

**DOI:** 10.3390/cells10030508

**Published:** 2021-02-27

**Authors:** Joao Pedro Garcia, Lizette Utomo, Imke Rudnik-Jansen, Jie Du, Nicolaas P.A. Zuithoff, Anita Krouwels, Gerjo J.V.M. van Osch, Laura B. Creemers

**Affiliations:** 1Department of Orthopedics, University Medical Centre Utrecht, 3584 CX Utrecht, The Netherlands; j.p.marquesgarcia-2@umcutrecht.nl (J.P.G.); jansen.imke@gmail.com (I.R.-J.); j.du-2@umcutrecht.nl (J.D.); A.Krouwels-2@umcutrecht.nl (A.K.); 2Department of Oral and Maxillofacial Surgery & Special Dental Care, University Medical Center Utrecht, 3584 CX Utrecht, The Netherlands; l.utomo-2@umcutrecht.nl; 3Department of Clinical Sciences, Faculty of Veterinary Medicine, Utrecht University, 3584 CL Utrecht, The Netherlands; 4Julius Center for Health Sciences and Primary Care, University Medical Center Utrecht, 3584 CX Utrecht, The Netherlands; NPAZuithoff@umcutrecht.nl; 5Department of Orthopaedic Surgery, Erasmus MC, University Medical Center Rotterdam, 3015 GD Rotterdam, The Netherlands; g.vanosch@erasmusmc.nl; 6Department of Otorhinolaryngology, Erasmus MC, University Medical Center Rotterdam, 3015 GD Rotterdam, The Netherlands

**Keywords:** osteoarthritis, oncostatin M, inflammation, biomarkers

## Abstract

Pro-inflammatory cytokines are considered to play a major role in osteoarthritis (OA), yet so far, the specific cytokines involved in the pathology of OA have not been identified. Oncostatin M (OSM) is a cytokine from the interleukin 6 (IL-6) family that has been shown to be elevated in synovial fluid of most rheumatoid arthritis (RA) patients, but only in a limited subset of OA patients. Little is known about OSM in the different joint tissues during OA and how its expression correlates with hallmarks of disease. Here, we mapped OSM expression in the joint tissues of two rat models of arthritis: an acute inflammatory model and an instability-induced osteoarthritic model. OSM expression was correlated with hallmarks of OA, namely cartilage damage, synovitis, and osteophyte formation. Reanalysis of an existing dataset on cytokine profiling of OA synovial fluid was performed to assess pattern differences between patients positive and negative for OSM. In the inflammatory model, OSM expression correlated with synovitis and osteophyte formation but not with cartilage damage. On the contrary, in the instability model of OA, an increase in synovitis, cartilage damage, and osteophyte formation was observed without changes in OSM expression. In line with these findings, synovial fluid of OA patients with detectable OSM contained higher levels of other inflammatory cytokines, namely interferon gamma (IFN-γ), IL-1α and tumor necrosis factor alpha (TNF-α), likely indicating a more inflammatory state. Taken together these data indicate OSM might play a prominent role in inflammatory phenotypes of OA.

## 1. Introduction

Osteoarthritis (OA) is the most prevalent musculoskeletal disorder and a leading cause of disability and morbidity worldwide [1,2]. It is a chronic and degenerative joint disease not only characterized by progressive cartilage degradation, but also by changes in subchondral bone, ligaments, synovial lining, and joint muscles [2]. Even though OA is known to be a complex disease with multiple etiologies, it is widely accepted that mechanical factors, together with both biochemical cues are the main key players in the disease pathogenesis [2,3]. Chondrocytes and synovial cells are thought to jointly act as catalyst for the degeneration process by producing matrix-degrading enzymes, such as metalloproteinases (MMPs) and aggrecanases (or a disintegrin and metalloproteinase with thrombospondin motifs—ADAMTSs) [2]. Contrary to rheumatoid arthritis (RA), OA is not clinically described as an inflammatory disease, although synovial inflammation, joint swelling, and pain are common findings in OA patients [3,4]. Tumor necrosis factor alpha (TNF-α) and interleukin 1 (IL-1) are two of the most studied mediators in OA. In vitro, they have been shown to increase production of matrix-degrading enzymes and other inflammatory mediators such as MMPs 1, 3, and 13, ADAMTSs 4 and 5, and IL-6 in chondrocytes [3]. However, growing evidence suggests that IL-1 and TNF-α do not play a major role in OA. Several clinical trials targeting these cytokines have shown no effect [5,6,7], contrasting the clear alleviation of symptoms in RA upon their inhibition. These observations may be in part explained by the fact that in the OA joint the levels of these mediators are lower than those of their natural inhibitors, such as IL-1 receptor antagonist and soluble TNF-α receptors [8,9].

Another cytokine that was shown to be associated with OA is oncostatin M (OSM). OSM, a cytokine from the IL-6 family, is known to be an important mediator of inflammation in RA, and has been found to be elevated in synovial fluid of RA patients [10,11]. In RA, OSM is thought to promote cartilage degradation and bone erosion, and augment inflammation [12,13]. Moreover, OSM has been shown to promote cartilage degradation both in vitro and in vivo, especially in combination with other cytokines such as IL-1α and TNF-α [12,14,15,16,17].

In OA, OSM levels were found to be detectable in synovial fluid in up to 30% of the patients [10,18,19,20]. In vitro inhibition of OSM in OA synovial fluid promoted anabolic and repair processes in osteoarthritic cartilage [19]. OSM, as well as other cytokines from the IL-6 cytokine family, have been extensively described to also have important roles in bone development and formation [21]. In fact, OSM was shown to mediate bone homeostasis, with studies showing increased periosteal bone apposition, and others reporting bone resorption upon intra-articular OSM overexpression [7,13,16,22,23]. Accordingly, OSM has been found in osteoblasts and bone lining cells [23]. Additionally, OSM has been shown to be produced by macrophages [11] and neutrophils [24] both in vitro and in vivo, in in vitro-activated T lymphocytes [25], and mast cells [26]. These cells are present in both the synovial lining and synovial fluid during inflammation in both OA and RA [27]. However, to what extent other cells and tissues in the joint produce OSM, and how this expression correlates with hallmarks of arthritis, such as cartilage damage, inflammation and osteophyte formation is still unclear.

The goal of this study was to map OSM expression in the joint in two different rat models: the peptidoglycan-polysaccharide-induced arthritis model (PGPS) and the anterior cruciate ligament transection and partial meniscectomy of the medial meniscus model (ACLT). While the first is an acute inflammation model, predominantly characterized by localized synovitis [28], the latter is characterized by joint destabilization and subsequent development of OA [29]. We also evaluated whether OSM expression is associated with hallmarks of disease, such as cartilage damage, synovial inflammation, and osteophyte formation in these models. Additionally, we analyzed the expression pattern of synovial cytokines in relation to expression of OSM in an existing dataset on cytokine expression in synovial fluid samples from OA patients.

## 2. Methods

### 2.1. Synovial Fluid Collection and Cytokine Measurements

Data of previously published cytokine profiles in synovial fluid of OA patients were re-analyzed [20]. In brief, synovial fluid was collected from 27 OA patients (mean age = 70; OSM− = 21; OSM+ = 6) undergoing total knee arthroplasty and according to the Medical Ethical regulations of the University Medical Centre Utrecht and the ‘good use of redundant tissue for clinical research’ guideline constructed by the Dutch Federation of Medical Research Societies on collection of redundant tissue for research. Cytokine measurements of OSM, IL-1α, IL-1β, IL-4, IL-6, IL-7, IL-8, IL-10, IL-13, TNF-α, IFN-γ, and IL-1Ra were measured using a multiplex bead assay (Luminex, Luminex Corporation, Austin, TX, USA), as described previously [20].

### 2.2. Animal Models

All animal experiments were carried out according to the ARRIVE (Animal Research: Reporting of In Vivo Experiments) guidelines and the approved protocol (AVD108002015282, approval date: 25 November 2015; WP#800-15-282-01-003 and WP#104970-2) of the Utrecht University Ethical Committee for Animal Care and Use, following the central commission of animal experiments guidelines for animal research in the Netherlands. Then, eight weeks old female Sprague Dawley rats of approximately 230 g (Charles River Laboratories, Amsterdam, The Netherlands) were used for this study. Rats were allowed to acclimatize for 1 week and housed in groups of 3 to 4 rats per cage which included enrichment under a 12 h light–dark cycle. Ad libitum food and water was provided.

In the PGPS model, 6 rats were randomly taken from their cages and local synovitis was induced by priming their left joint (experimental joint) by an intra-articular injection of 25 µL PGPS (100 P fraction with 5 mg rhamnose/mL PGPS; Lee Laboratories, Grayson, GA, USA) at a concentration of 0.17 mg/mL, under general isoflurane anesthesia (day 14) [30]. At days 0, 14, and 28 synovitis was reactivated in the experimental knee joints by intravenous injection of 500 µL of PGPS (0.28 mg/mL PGPS) via the tail vein. Animals were euthanized after 6 weeks with CO_2_. For the ACLT model, OA was induced unilaterally through anterior cruciate ligament transection and partial medial meniscectomy in the left knee of 6 rats that were randomly taken from their cages, under general isoflurane anesthesia [31]. After 16 weeks, rats were euthanized with CO_2_.

### 2.3. Histology

After euthanasia, all rat hind limbs were collected for histological analysis and joints were fixed in 4% formaldehyde solution (VWR international BV, Amsterdam, The Netherlands) at room temperature for 1 week and subsequently decalcified at room temperature in 0.5 M ethylenediaminetetraacetic acid (EDTA; VWR international BV, Amsterdam, The Netherlands) solution for a total of 8 weeks. Re-fixation was performed for 3 days in 4% formaldehyde every 2 weeks. Then, samples were embedded in paraffin and cut into 5-µm thick coronal knee joint sections. Sections were taken from the middle part of the joint. Samples from the ACLT model were not sufficiently decalcified and had to be deparaffinized, decalcified in EDTA for an additional 4 weeks and re-embedded in paraffin.

Sections were stained with safranin-O/fast green to evaluate cartilage damage using the Mankin score (0 complete healthy—14 total cartilage destruction) [32]. Hematoxylin/eosin staining was used to evaluate synovitis using the Krenn score [33]. Scoring was done in a random order by two independent observers (JPG, IRJ) blinded for treatment, and scores were averaged.

Staining for tartrate-resistant acid phosphatase (TRAP) activity was performed to detect osteoclasts. Sections were deparaffinized in xylene, rehydrated, washed in running tap water for 5 min, and incubated with 0.2 M acetate buffer-tartaric acid for 20 min at 37 °C. Subsequently, Naphtol AS-MX phosphate (0.5 mg/mL; Sigma–Aldrich, St Louis, MA, USA) and Fast red TR salt (1.1 mg/mL; Sigma–Aldrich, St Louis, MA, USA) were added to the samples and incubated for another 3 hours. Sections were counterstained with Mayer’s hematoxylin. Osteoclasts were defined as multinucleated TRAP-positive cells in the periosteum, growth plate and subchondral bone of both tibia and femur. The number and brightness of stained osteoclasts were assessed and scored on a scale from 1 to 5 (0 = no staining, and 4 = numerous positive cells and bright staining). Scoring was done in a random order by two independent observers (JPG, AK) blinded for treatment, and scores were averaged.

### 2.4. OSM Immunohistochemistry

Sections were deparaffinized in xylene, rehydrated, and subsequently blocked for nonspecific endogenous peroxidase by incubating with 0.3% H_2_O_2_ for 10 min. Due to the re-embedding step in the ACLT joints, antigen retrieval was performed for these samples by incubating them with 1 mg/mL pepsin (Sigma–Aldrich, St Louis, MA, USA) for 2 h at 37 °C followed by incubation with 10 mg/mL hyaluronidase (Sigma–Aldrich, St Louis, MA, USA for 30 min at 37 °C. Between each step, slides were washed three times with phosphate buffered saline (PBS) containing 0.1% Tween-20 (PBS-T) for 5 min. We confirmed that antigen retrieval did not affect OSM staining in the PGPS model, nor was relative staining intensity between the tissues changed, suggesting staining patterns were not dependent on tissue processing or antigen retrieval in the two models (Appendix A). Then, sections were blocked in PBS containing 5% bovine serum albumin for 30 min followed by overnight incubation at 4 °C with the primary antibody (2 µg/mL mouse monoclonal OSM (B-6), sc-133229; Santa Cruz Biotechnology, Dallas, TX, USA). Mouse IgG isotype at 2 µg/mL was used as negative control (Appendix A). The next day, sections were washed three times with PBS-T for 5 min, followed by incubation with the secondary antibody BrightVision poly-HRP-anti mouse for 1 h at room temperature (Immunologic, Duiven, The Netherlands). After washing with PBS-T, sections were incubated with 3,3’Diaminobenzidine (DAB) substrate for 5–10 min, briefly counterstained with Weigert’s Hematoxylin and rinsed with running tap water for 10 min. Finally, sections were dehydrated with a series ethanol and xylene and permanently mounted with Depex (Sigma–Aldrich, St Louis, MA, USA). Specificity of the B-6 anti-OSM antibody was confirmed by western blot analysis on primary rat chondrocytes and PC-12 cells, a rat cell line from adrenal medulla, using recombinant rat OSM (400-36, PeproTech, London, UK) as control. The expression patterns of these samples were also compared with a second anti-OSM antibody (clone A-9, sc-374039; Santa Cruz Biotechnology, Dallas, MA, USA). Cells were harvested using RIPA lysis buffer (ab156034; Abcam, Cambridge, UK) containing protease inhibitors (cOmplete™, Mini Protease Inhibitor Cocktail, 11836153001; Merck, Amsterdam, The Netherlands). Total protein content was determined with a Pierce BCA Protein assay (23225; ThermoFisher Scientific, Waltham, MA, USA) according to manufacturer’s instructions. For cell lysates, 100 µg total protein were used, while for recombinant OSM a total amount of 20 and 40 ng was used. The protein samples were separated by dodecyl sulphate polyacrylamide gel electrophoresis (SDS-PAGE) and transferred to Nitrocellulose membranes (0.45 μm, 1620115; Bio-Rad Laboratories, Hercules, CA, USA). Following blocking with 5% non-fat dry milk in PBS, the membranes were incubated overnight at 4 °C with anti-OSM (B-6) and anti-OSM (A-9) (final concentration of 2 µg/mL in 5% non-fat dry milk). Following washing, membranes were incubated with polyclonal rabbit anti-mouse-HRP (0.1 µg/ML, P0260; Agilent Technologies, Santa Clara, CA, USA) for 1 h at RT. Finally, membranes were incubated with SuperSignal West Femto Maximum Sensitivity Substrate (340796; ThermoFischer Scientific, Waltham, MA, USA) and imaged with a ChemiDoc^®^ Touch Imaging system (Bio-Rad Laboratories, Hercules, CA, USA). Image analysis was performed Image Lab Software (version 6.0.1; Bio-Rad Laboratories, Hercules, CA, USA). Similar protein bands were detected using both antibodies in both cell extracts, confirming the specificity of the used antibody for immunohistochemistry (Appendix A).

For quantification purposes, the stained joint sections were ranked from 0 to 3 (0 = no staining; 3 = maximum staining) based on OSM staining intensity in the articular cartilage, periosteum, menisci, femorotibial synovium, and collateral ligaments. Cartilage, periosteum, and menisci were further divided in individual compartments (Appendix A). Scoring was performed using an optical microscope in a random order by two independent observers blinded for treatment (JPG, LU), and scores were averaged. Pictures presented in the manuscript serve as representative images.

### 2.5. Micro-CT Analysis

Directly after euthanasia, animals were imaged using a Quantum FX µ-CT scanner (PerkinElmer, Waltham, MA, USA) with the following parameters: time = 3 min, isotropic voxel size = 30 µm^3^, tube voltage of 90 kV, tube current = 180 µA. Scans were reconstructed using the scanner’s software (PerkinElmer, Waltham, MA, USA). ImageJ software was used for image analysis (ImageJ, 1.47v, NIH, Bethesda, ML, USA). Serial 2D scans of the femur, tibia, and patella were evaluated for subchondral sclerosis and osteophyte volume.

### 2.6. Statistics

IBM SPSS 25.0 (IBM, New York, NY, USA) was used for statistical analysis. The intra-class correlation coefficient (ICC) was used to calculate inter-rater variability for the OSM scoring system. A two-way mixed-effect model based on a mean-rating (*k* = 2) and absolute agreement was used, as previously described [34]. ICC estimates and the 95% confidence intervals (CI) were reported.

Non-parametric Kruskal–Wallis tests were conducted to evaluate statistically significant differences between treatment groups (control vs. induced) and between compartments (i.e., femur vs. tibia, and lateral vs. medial). The correlation between cartilage damage, synovitis and OSM staining was evaluated with a non-parametric Spearman rho correlation test. *p*-values of <0.05 were considered statistically significant.

Fischer exact test was performed to examine the relation between frequency of osteophyte formation and induction of OA. Osteophyte presence was classified as Yes or No for each of the four analyzed compartments (lateral and medial femur condyles, and lateral and medial tibial plateaus) for a total of 24 compartments (4 compartments × 6 joints). Chi-square statistics are reported with degrees of freedom and sample size, the Pearson chi-square value and the significance level.

On the dataset of cytokine levels in synovial fluid of OA patients, Spearman’s rank correlation was used to analyze the relationship between OSM expression and expression of the remaining cytokines in the human synovial fluid. To analyze differences in cytokine expression between patients with detectable and non-detectable levels of OSM (OSM+ and OSM−), and due to the small number of patients, a generalized linear model was used. The generalized linear model was performed with an unstructured (i.e., GEE type) residual covariance matrix to analyze multiple normally distributed outcomes. Since the distribution of cytokine concentration values were highly skewed, we used log transformed outcomes for the statistical analysis. We specifically choose this model to obtain a single likelihood ratio test for the hypothesis that the pattern of cytokine expressions differs between patient groups (OSM− vs. OSM+). Additionally, the model estimated differences between patient groups for each cytokine separately. For ease of interpretation, these differences (with 95% CI) were transformed back to the original scale by taking the exponential and applying a smearing factor [35]. The *p*-values of the Wald tests for individual cytokine expressions were used to rank the differences between groups.

## 3. Results

### 3.1. Mapping and Quantification of OSM Expression in Joint Tissues

To map the presence of OSM in different phenotypes of joint diseases, OSM expression was evaluated by immunohistochemistry in two mouse models: inflammatory arthritis (PGPS model) and degenerative OA (ACLT) model. Specificity of the staining was confirmed by the absence of staining in the negative control, performed with IgG isotype (Appendix A). In the PGPS model, extracellular OSM staining was observed in articular cartilage, meniscus, periosteum, and synovium (Figure 1). Additionally, intracellular staining was observed in chondrocytes and meniscal fibrochondrocytes (Figure 2 and Appendix A). In both tissue structures, OSM-positive cells were found in the superficial cell layers, while absent in deeper layers. In the synovium, intracellular OSM staining was prominent in the subintimal layers rather than the lining. In the ACLT model, extracellular OSM staining was found in the periosteum and synovium, but not in cartilage or meniscus (Figure 1). Even though synovial staining was less extensive, intracellular staining was observed in the lining (Figure 2). Similar to the PGPS model, intracellular staining was found in articular cartilage and menisci.

Expression of OSM was semi-quantified by scoring the knee tissues from 0 to 3 based on OSM staining intensity in the articular cartilage, periosteum, menisci, synovium, and ligaments. The ICC values for inter-rater variability were 0.828 (95% CI, 0.766–0.875) for the PGPS model and 0.827 (95% CI, 0.758–0.873) k for the ACLT model, indicating a good reliability among the two independent scorers.

In the PGPS model, OSM staining was more predominant in cartilage of the lateral femoral condyle, and in both lateral and medial tibial plateau cartilages of the diseased knees as compared to control knees, but this was not the case in the ACLT model (Figure 3A). Regarding the femoral periosteum, OSM expression in lateral and medial compartments of diseased joints was higher than control joints in the PGPS model, but not in the ACLT model (Figure 3B).

Expression of OSM in the tibial periosteum was not affected by induction of arthritis in any of the models. No differences in OSM expression were observed in the lateral or medial meniscus in neither of the models (Figure 3C). Finally, expression was higher in the synovium of PGPS-injected joints than in the synovium of the control joints (Figure 3D). No differences in OSM expression were found in the ligaments in the PGPS knees, and in the synovium and ligaments in the ACLT joints.

Additionally, in both models, OSM expression in the tibial periosteum was significantly higher in the medial compartment compared to lateral, independently of induction of disease (Figure 3B). This was not observed for the femoral periosteum.

### 3.2. Expression of OSM Correlates with Synovitis in the PGPS Model of Acute Arthritis

While the overall OSM score was shown to be significantly higher in PGPS-injected knees than in control knees (*p* < 0.01; Figure 4A), no differences were observed between OA or control knees in the ACLT model (*p* = 0.873).

In both models, induction of disease led to increased synovitis according to the Krenn score (Figure 4B), however cartilage damage was only observed in induced joints of the ACLT model (*p* < 0.05) (Figure 4C).

Furthermore, the total OSM score strongly correlated with synovitis in the PGPS model (ρ = 0.848, *p* = 0.001, Figure 4D), but not in the ACLT model (ρ = 0.272, *p* = 0.387, Figure 4D). No correlation was found between OSM expression and cartilage damage in either model (Figure 4E).

Finally, osteophyte volume was moderately correlated with the total OSM expression in the PGPS knees (ρ = 0.623, *p* = 0.034), but not in the ACLT knees (ρ = 0.238, *p* = 0.472) (Figure 4F). No differences in subchondral sclerosis were observed between induced and control joints in either model (Appendix A). Likewise, no differences in TRAP staining were found between induced and control joints in neither model (Appendix A).

### 3.3. Cytokine Profile of Osteoarthritis Patients

OSM was detected in the synovial fluid of 6 out of 27 patients (OSM+), and was undetectable in 21 patients (OSM−). The concentration of OSM showed a positive and significant correlation with the concentration of other pro-inflammatory cytokines, namely IL-1α (ρ = 0.461, *p* = 0.016), TNF-α (ρ = 0.481, *p* = 0.011), and IFN-γ (ρ = 0.573, *p* = 0.002) (Appendix A). A generalized linear mixed model was conducted with and without explanatory variables (OSM grouping) to assess differences in cytokine expression pattern between OSM+ and OSM− patients. A likelihood ratio test between the two models indicated significant difference in cytokine expression patterns between the two patient groups (χ^2^ (11) = 43.48, *p* < 0.0001). The lowest *p* values between patient groups were observed for pro-inflammatory cytokines (IFN-γ, IL-1α, and TNF-α), while no significant differences were observed for any of the anti-inflammatory cytokines (Figure 5 and Appendix A).

## 4. Discussion

In this study, we have, for the first time, mapped the presence of OSM in various joint tissues in two models of arthritis. OSM expression correlated with disease hallmarks in the acute inflammatory arthritis model, but not in the instability-induced osteoarthritis model, suggesting an association with a more inflammatory phenotype. This was in line with the reanalysis of previously obtained data in a patient cohort, where we observed that OSM concentration positively correlated with other inflammatory cytokines, namely IL-1α, TNF-α, and IFN-γ. Furthermore, generalized linear model analysis showed these cytokines to be the most likely cytokines to be differentially expressed among the patient group with and the patient group without OSM expression.

Extensive extracellular OSM expression was recognized in the synovial capsule, possibly derived from infiltrating inflammatory cells such as macrophages, neutrophils, and T-cells, as previously reported [11,24,25]. These cell types are known to migrate to synovium during OA and RA, and to mediate inflammatory pathways [27]. High neutrophil counts are detected in most RA patients compared to healthy controls [37], while in OA patients, neutrophil presence is associated only with severe and late stages [38]. In line with this, cell infiltration, synovitis, and synovial OSM staining were indeed seen to a greater extent in the inflammatory (PGPS) mouse model.

Intracellular OSM staining was observed in the most superficial layers in articular cartilage and meniscus. In human OA patients, OSM levels in cartilage tissue were previously shown to be even higher than in synovial fluid [20]. However, to what extent chondrocytes contribute to the disease process is still unclear.

Interestingly, OSM staining was also found in the tibial and femoral periosteum. OSM has been previously detected in osteoblasts, bone-lining cells, and osteocytes [23]. Intra-articular overexpression of OSM has been shown to promote periosteal bone formation and deposition [22,23], in a process that is thought to occur through inhibition of sclerostin [23,39], a negative regulator of bone formation. However, OSM has also been associated with bone erosion when simultaneously overexpressed with IL-1β or TNF-α [13,16,17]. Here, the presence of OSM in the periosteum of both models indicates this cytokine may be part of the as yet enigmatic role of bone homeostasis and remodeling in OA [2,40]. Additionally, OSM expression was higher in the medial tibial periosteum than in the lateral, independent of induction of arthritis. Gait analysis in mice has shown weight is distributed more towards the medial side than the lateral, and mechanical loading has been shown to enhance OSM gene expression [35,41,42].

OSM expression correlated with osteophyte volume in the PGPS model, but not in the ACLT model. While the exact mechanisms leading to osteophyte formation are still unknown, osteophyte formation was shown to be associated with synovial inflammation and macrophage infiltration [43,44]. Hence, one could hypothesize that OSM production by infiltrating cells could potentially contribute to osteophyte formation in a direct or indirect manner in the PGPS model but not in the traumatic ACLT model.

Furthermore, OSM expression also correlated with the degree of inflammation in the PGPS model, but not in the ACLT model despite increased synovitis. This might suggest OSM to play a more prominent role in inflammation of the PGPS model and to associate with a more inflammatory phenotype. These findings are supported by analysis of OA synovial fluid cytokines, which showed that OSM concentration correlated with other pro-inflammatory cytokines, in particular IFN-γ, IL-1α, and TNF-α. Additionally, these cytokines were shown to be elevated in OSM+ patients.

Synovial levels of OSM were shown to be higher in patients with cartilage defects than in healthy subjects [20]. However, in this study, cartilage damage was only found in the ACLT model and not in the PGPS model. This may be due to the fact that in PGPS-induced arthritis animals were sacrificed after 6 weeks compared to 16 weeks in the ACLT model. Nevertheless, also in a long-term PGPS study after 12 weeks no cartilage damage was observed, despite increased synovitis [45]. Since no correlation was found between cartilage damage and OSM score in either of the models, it can be hypothesized that in late-stage disease, damage is a cause rather than a consequence of inflammation. Possibly, inflammatory cytokines play a role in the early chain of events initiating rather than maintaining OA. TNF-α a and IL-1β, for instance, have been shown to be still elevated 12 weeks after ACLT in a rat model [46]. Additionally, it was shown before that synovial levels of IL-1β were up-regulated immediately upon anterior cruciate ligament (ACL) injury in humans and remained elevated more than 3 months after injury, while TNF-α levels progressively increased after injury [47]. TNF-α and IL-1β levels post trauma were also shown to be associated with severity of cartilage damage upon ACL injury in patients [48], and inhibition of IL-1R immediately after injury was suggested to improve joint integrity [49].

Although we did not evaluate if other cytokines were elevated in the PGPS model, it is known that OSM alone at the concentrations found in the OA synovial fluid does not promote cartilage degradation in vitro, but may rather inhibit matrix production [19]. On the other hand, even though OSM expression was not elevated in the ACLT model 16 weeks post induction, it is still possible that low levels of OSM are a prerequisite to act in concert with other pro-inflammatory cytokines to promote cartilage damage. Additionally, we cannot exclude that OSM levels were elevated at earlier time points, as a consequence of the traumatic injury.

One of the limitations of this study is the fact that OSM expression in the OA models was evaluated by immunohistochemistry at the endpoint of the experiment. Additionally, the study would benefit from having midpoint histological analysis to assess the presence of OSM and the extent of cartilage damage and synovitis over time. Moreover, we cannot exclude the intracellular staining observed in the chondrocytes is produced by the chondrocytes or derived from other cells. We can conclude however that OSM was produced in the joint tissues in both models, though we did not find differences between models. Another limitation of the study is the small patient number available for analysis of cytokine expression in OA synovial fluid, as a higher number of patients would increase the power of the statistical analysis. Additionally, the study would benefit from information regarding the severity of OA (i.e., Kellgren–Lawrence score, Mankin score) and synovitis (i.e., Krenn score), and patient scores. Thus, a more complete comparison could be done between the clinical data and the data obtained with the experimental models.

In conclusion, in this study we have shown that OSM expression is associated with synovial inflammation and osteophyte formation, but not with cartilage damage in an acute arthritis model. On the other hand, although the ACLT model resulted in cartilage damage, synovial inflammation, and osteophytes, these parameters did not correlate with OSM expression. Additionally, in a cohort of OA patients, patients with detectable OSM had higher expression of other pro-inflammatory cytokines. Taken together, these data suggest that OSM might have a prominent role in inflammatory types of OA, and can potentially be used in a panel of biomarkers for patient stratification or as a target for future drug development. To further elucidate the role of OSM in OA, future studies should focus on in vivo inhibition of OSM upon induction of OA, and on the evaluation of the subsequent effect on cartilage damage, inflammation, and other hallmarks of arthritis.

## Figures and Tables

**Figure 1 cells-10-00508-f001:**
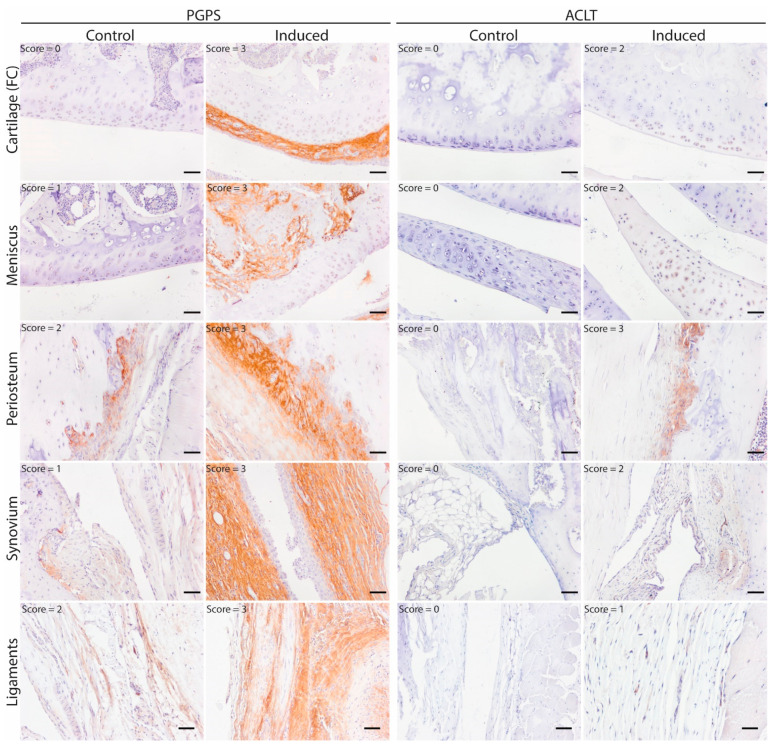
OSM expression pattern in acute arthritis and instability-induced OA. Representative pictures of OSM expression (brown staining) in different joint structures: Femur condyle (FC) cartilage (chondrocytes), meniscus, periosteum and synovium. Acute arthritis in the PGPS model was induced by intra-articular injection of PGPS in the left knee. The ACLT model was induced unilaterally (left knee) through anterior cruciate ligament transection and partial medial meniscectomy. Individual scores for each structure are displayed in the top left corner of each picture. Scale bar: 50 µm.

**Figure 2 cells-10-00508-f002:**
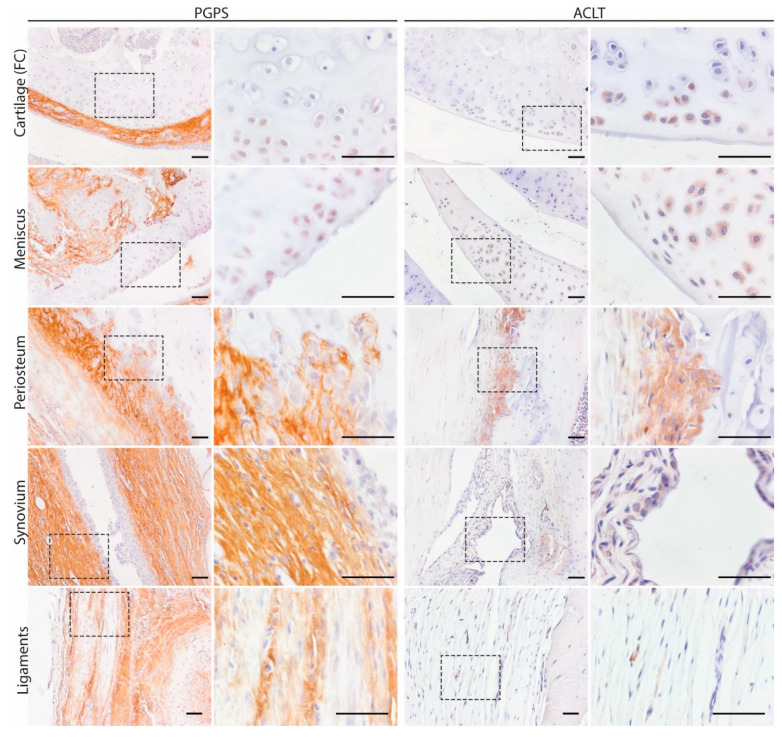
Intra and extracellular OSM expression patterns in acute arthritis and instability induced OA. Representative pictures of OSM expression in different joint structures: Femur condyle (FC) cartilage (chondrocytes, meniscus, periosteum, synovium, and ligaments. Scale bar: 50 µm.

**Figure 3 cells-10-00508-f003:**
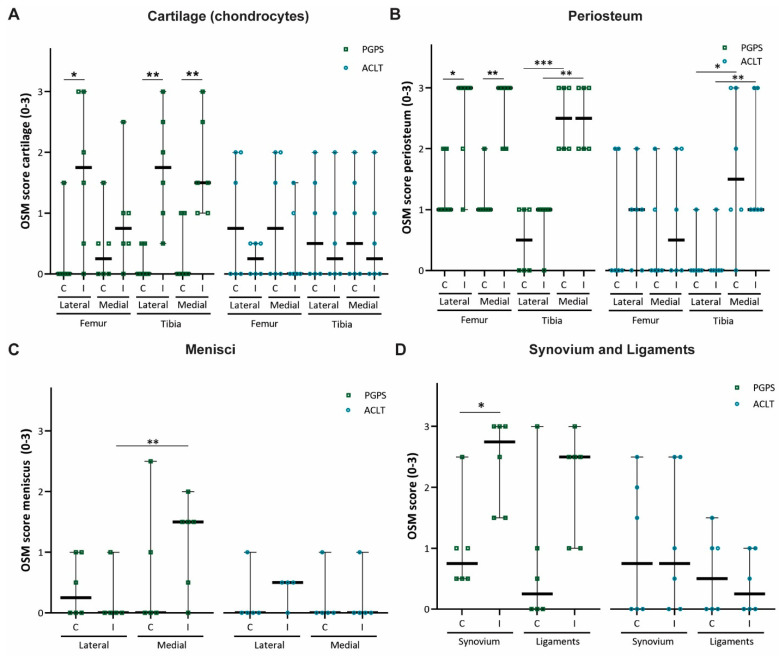
OSM expression scores in two rodent models of acute arthritis and instability induced OA. OSM score per anatomical structure: cartilage (chondrocytes) (**A**), periosteum (**B**), meniscus (**C**), synovium, and ligaments (**D**). Control (C), Induced (I). 0 = no staining, 1 = slight staining, 2 = moderate staining, 3 = extensive staining. Each individual datum point represents a rat knee. Squared symbols (green) refer to the PGPS model. Circular symbols (blue) refer to the ACLT model. Data are presented as median and 95% CI. (* *p* < 0.05, ** *p* < 0.01, and *** *p* < 0.001).

**Figure 4 cells-10-00508-f004:**
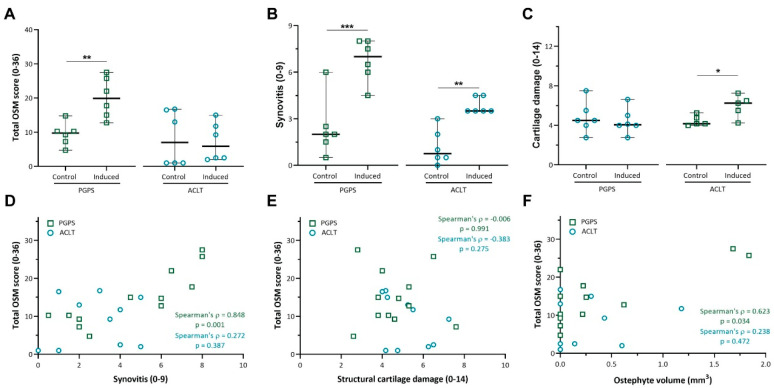
Total OSM expression and correlation with synovitis, cartilage damage and osteophyte volume. (**A**) Total OSM score was calculated by adding up the individual scores for every compartment (min = 0; max = 32). (**B**) Synovitis was scored according to the Krenn score. (**C**) Cartilage damage was scored according to the Mankin score. Data are presented as median and 95% CI. Each individual datum point represents a rat knee. (*: *p* < 0.05, **: *p* < 0.01, and ***: *p* < 0.001). (**D**) Correlation scatterplot between total OSM staining and synovitis score in both PGPS and ACLT models. (**E**) Correlation scatterplot between total OSM staining and articular cartilage damage in both PGPS and ACLT models. (**F**) Correlation scatterplot between total OSM and osteophyte volume in both PGPS and ACLT models. Each individual datum point represents a rat knee.

**Figure 5 cells-10-00508-f005:**
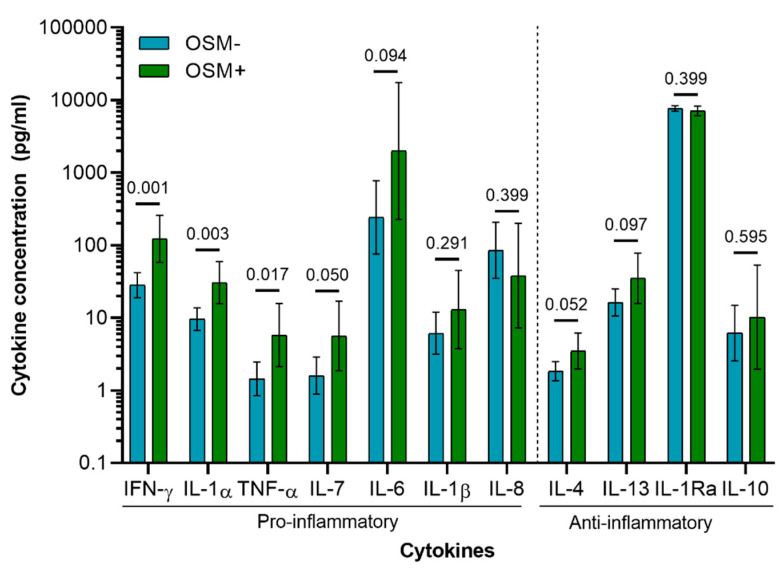
Cytokine concentration in synovial fluid of OA patients (*n* = 27; OSM− = 21; OSM+ = 6). Concentration values were back transformed to the original scale as described previously [36]. Values presented are the estimated means and 95% CI derived from the general linear mixed model, with the *p*-values of the Wald tests performed between patient groups (OSM− and OSM+) for each cytokine.

## Data Availability

All data associated with this study are available in the main text or the Appendix A.

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
