# Peer review of "Association between Oncostatin M Expression and Inflammatory Phenotype in Experimental Arthritis Models and Osteoarthritis Patients"

_cells, 2021, doi:10.3390/cells10030508_

Round 1

Reviewer 1 Report

The results:
Figures 1 and 2 they are not very clear. They need to have more resolution to better observe the markings. I recommend for authors to place a figure showing initially the histology of all regions used in the markings (H; E staining; Hematoxylin and Eosin). These findings will serve as a basis for locating the changes caused by digestive models of Arthritis and the main inflammatory cells that will be involved in the production of cytokines. The scores should be reviewed, as in some regions the markings are not clear (Figures 3 and 4).
Figure 5- Has well presents problems.
The way of presenting these cytokine data is not adequate. Authors should compare proinflammatory cytokines with their respective controls and not mix proinflammatory and antiinflammatory drugs in the same graph, as they tend to have different expressions in each group. The scatter plot are the most suitable.
The conclusions should again be placed in line with the changes and suggested results.

Reviewer 2 Report

In this manuscript, Garcia et al mapped the expression of OSM in the joints of 2 different rat models (PGPS and ACLT). They found that OSM expression correlated with synovial inflammation and osteophyte formation in the acute arthritis model. At the opposite, they did not found correlation in the ACLT model. No correlation was found between OSM expression and cartilage damage in either model. Finally, they analyzed OSM expression in the SF of patients with OA, and they found a positive correlation with pro-inflammatory cytokines. They concluded that OSM might have a prominent role during inflammatory phases of OA.

The manuscript is clearly written, and interesting to read. The limitations of the study (especially the number of human samples and the lack of notion of kinetic of OSM expression in the animal models) are also well discussed.

Minor comment:

Fig.5: please indicate in the legend the number of OA patients tested.

Author Response

This manuscript is a resubmission of an earlier submission. The following is a list of the peer review reports and author responses from that submission.

Round 1

Reviewer 1 Report

This is a descriptive study examining the expression of OSM in two different models of joint destruction in rats (and thus new results) and the presence of OSM in synovial fluid (SF) of OA patients. The study uses IHC with a mouse monoclonal antibody to assess presence in fixed rat tissues, and specific ELISAs to examine soluble OSM and other cytokines in human SF.

The following issues need to be addressed.

1) Use of antibodies in IHC require careful analysis of specificity. The study uses a commercially available mouse monoclonal against rat OSM. The use of IgG isotype as a comparator controls nicely for potential secondary Ab effects. However, the specificity of the primary anti-OSM Ab is assumed in the manuscript. It is not clear if the monoclonal picks up only OSM using the methods necessary to get IHC signal. It would be important to confirm specificity in the rat joint tissues (this antibody works in western blot analysis, and could be used on homogenized rat tissues for example) to strengthen the assumption that the staining is OSM-specific. Can exogenous rat OSM ligand added to the Ab block the staining? (Staining of OSM-KO tissues would be even more definitive but may not be available).

2) The assertions of intracellular staining in some cells (chondrocytes for example, fig 2) is not clear. IHC will not easily determine surface vs intracellular staining. This needs to be addressed. Are the chondrocyte-positive cells (fig 3) producing OSM or localizing it? It was not clear if such chondrocyte staining was different between the induced models. (Figure 2 photos would seem to suggest they are similar). Does fig 3A refer to chondrocyte-localized staining in the models? Chondrocytes have surface receptors for OSM. If the staining is specific, and the cells are actually producing OSM in both models, what are the ramifications?

3) The interpretation that OSM is elevated in a highly inflammatory environment (PGPS) but not in a lower inflammatory environment (ACLT) is not surprising on the basis of previous literature as discussed by the authors. The association of elevated OSM with elevation of other cytokines in the patient cohort would benefit from data regarding the inflammatory status of the OSM- vs the OSM+ patient groups. Is OSM a better marker than any of the other cytokines assessed to be potentially used for patient stratification?

Reviewer 2 Report

Minor revisor :

The present work had as one of the objectives to map the presence of OSM in different phenotypes of joint diseases, the expression of OSM was evaluated by immunohistochemistry in two mouse models: inflammatory arthritis (PGPS model) and degenerative OA model (ACLT). Although the results were good, the authors do not characterize the main cells that produce OSM and that these cells could be modulating the cytokine profile in the two different models. The characterization of these cells is very  importante to investigate the main mechanisms involved in arthritis models.